# A Capillary Electrophoresis Method Based on Molecularly Imprinted Solid-Phase Extraction for Selective and Sensitive Detection of Histamine in Foods

**DOI:** 10.3390/molecules27206987

**Published:** 2022-10-17

**Authors:** Yixuan Fan, Runze Yu, Yongfeng Chen, Yufeng Sun, Geoffrey I. N. Waterhouse, Zhixiang Xu

**Affiliations:** 1Key Laboratory of Food Processing Technology and Quality Control in Shandong Province, College of Food Science and Engineering, Shandong Agricultural University, Tai’an 271018, China; 2School of Chemical Sciences, The University of Auckland, Auckland 1142, New Zealand

**Keywords:** histamine, molecular imprinting technology, solid-phase extraction, capillary electrophoresis

## Abstract

In this study, a sensitive capillary electrophoresis (CE) method based on molecularly imprinted solid-phase extraction (MISPE) was proposed to determine histamine in foods. A molecularly imprinted polymer (MIP) synthesized by bulk polymerization was used as the MISPE adsorbent for the selective extraction of histamine. Under the optimal conditions, the MISPE-CE method possessed good linearity for histamine detection in the concentration range of 0.1–100.0 μg/L. The limit of detection and limit of quantification of the method were calculated to be 0.087 μg/L and 0.29 μg/L, respectively. The histamine in spiked rice vinegar and liquor samples were detected by the developed method with recoveries of 92.63–111.00%. The histamine contents in fish, prawn, pork, chicken breast and soy sauce samples were determined using the developed method and a high-performance liquid chromatography method, with no significant difference found between the two methods.

## 1. Introduction

Histamine (C_5_H_9_N_3_), a common biological amine, is produced by the action of the enzyme histidine decarboxylase or histamine-producing bacteria (HPB) on histidine [1]. Histamine is frequently found in aquatic products, brewed condiments and fermented meats [2,3]. Excessive intake of histamine may lead to health problems such as headache, high blood pressure and even death [4,5]. The maximum limit of histamine in imported aquatic products was set at 50 mg/kg by the United States Food and Drug Administration [6]. Therefore, the estimation of histamine levels in foods is receiving widespread attention.

To date, several analytical methods including enzyme-linked immunosorbent assay (ELISA) [7], high performance liquid chromatography (HPLC) [8,9], and gas chromatography (GC) [10] have been developed for histamine determination. Although the ELISA method offers a sensitive means of detection, it usually requires a long analysis time and the antibodies used are generally unstable [11]. In contrast, HPLC and GC methods have been widely used for the detection of histamine, but a complex derivatization process is often needed [12]. Capillary electrophoresis (CE) with rapid separation speed and short analysis times has been successfully applied in the fields of food science, analytical chemistry and pharmacy [13,14,15]. However, the sensitivity of CE is limited and a pre-treatment procedure is usually required.

In recent years, some pre-treatments such as liquid–liquid extraction, solid-phase extraction (SPE) and magnetic solid-phase extraction have been proposed to sped up CE analyses [16,17,18]. SPE is the most widely used among these procedures [19]. Though SPE greatly improves the sensitivity of CE, poor selectivity is often encountered when analyzing analytes in complex food matrices. Thus, preparing a selective material as the SPE sorbent is attracting significant research interest. The molecular imprinting technique is a promising strategy, involving the preparation of a molecularly imprinted polymer (MIP) with specific binding sites [20,21,22]. The synthesis and application of histamine-imprinted MIPs has been reported by several groups. For example, Sahebnasagh et al. prepared a series of MIPs and selected the optimized MIP as MISPE sorbent for the enrichment of histamine in canned fish [23]. Hashemi et al. demonstrated a chitosan MIP coated on Fe_3_O_4_ magnetic nanoparticles as the SPE sorbent for the selective detection of histamine in tuna fish [24]. The combination of SPE and MIP is a desirable option to overcome the lack of selectivity [25,26]. More importantly, the MISPE cartridges can be reused [27]. To date, several studies coupling analytical methods with MISPE have been reported. Dourado et al. explored the preparation of MIPs with high affinity, adsorption capacity and selectivity as MISPE adsorbents for saccharin determination in diet tea samples [28]. Ahmadi et al. synthesized MIPs of 1,8-cineole on hydroxyl-functionalized multiwall carbon nanotubes for the selective extraction of 1,8-cineole in water distillates of Artemisia *sieberi* (*sagebrush*) and *thyme* samples [29]. Hence, CE using MISPE as the sample pretreatment has great potential for the sensitive analysis of histamine in complex food systems.

In this paper, a MIP with high selectivity were prepared by bulk polymerization and used as the MISPE sorbent for extraction of histamine. A sensitive method of CE coupling with MISPE was proposed. The factors affecting CE separation (the types of running buffer, the pH of running buffer, the concentration of running buffer and the CE separation voltage) and MISPE enrichment (the eluent ratio and the eluent volume) were investigated. Moreover, the repeatability, selectivity, anti-interference capability, accuracy and applicability of the MISPE-CE method were evaluated. To the best of our knowledge, no previous reports have combined MISPE with the CE method to selectively and sensitively determine histamine in foods.

## 2. Results and Discussion

### 2.1. Synthesis and Characterization of the MIP

Histamine dihydrochloride, a non-imprinted polymer (NIP), and the histamine-imprinted MIP before elution and after elution were characterized by FT-IR. As shown in Figure 1, the NIP and MIP showed and absorption peaks at 3450 cm^−1^ (O-H), 2957 cm^−1^ (C-H), 1727 cm^−1^ (C=O), 1257 cm^−1^ (C-O) and 1159 cm^−1^ (C-O-C) due to the MAA polymer [30]. Histamine dihydrochloride and the MIP before elution showed additional intense absorption peaks at 617 cm^−1^ and 622 cm^−1^, respectively, associated with histamine [31]. The small shift in the absorption peak position for the MIP before elution was probably caused by the hydrogen bonding between the -NH_2_ groups of histamine dihydrochloride and the COO- of the MAA polymer. These results showed that a histamine MIP was successfully synthesized. After elution, the absorption peak at 622 cm^−1^ disappeared, confirming that the template molecules had been completely removed.

### 2.2. Optimization of the MISPE Procedures

In order to completely desorb the histamine bound to the MISPE cartridges, the MISPE procedure needed to be optimized [32].

The eluent is one of the key factors during a MISPE extraction process [33]. Hence, eluent with different ratios of methanol/acetic acid (100:0, 95:5, 90:10 and 80:20, *v*/*v*) were trialed. Figure 2a shows the effect of different ratios of methanol/acetic acid on the CE peak area. The highest peak area was obtained when the ratio was 90:10. Therefore, methanol/acetic acid (90:10, *v*/*v*) was selected as the eluent for further studies.

The volume of eluent was further investigated. Figure 2b indicates that the CE peak area increased with the eluent volume from 6.0 mL to 8.0 mL and then remained almost consistent at higher eluent volumes. Hence, 8.0 mL of methanol/acetic acid (90:10, *v*/*v*) was selected as the optimal eluent volume.

### 2.3. Optimization of the CE Conditions

In order to improve the sensitivity and separation of the histamine assay, the CE conditions including the types of running buffer, pH of running buffer, concentration of running buffer and the CE separation voltage required optimization.

The types of running buffer directly influenced the separation result and migration time. The separation of histamine in different running buffers (100 mmol/L) containing 1.0 mg/L histamine at 10 kV applied voltage is shown in Figure 2c. The NaH_2_PO_4_ (HCl) running buffer gave short migration times, a large peak area and a symmetrical peak shape for histamine. Therefore, NaH_2_PO_4_ (HCl) was used as the running buffer in following experiments.

The effect of the pH (2.0–4.5) of the running buffer (100 mmol/L) at 10 kV applied voltage on the separation of 1.0 mg/L histamine was next studied. As shown in Figure 2d, the resolution of histamine improved as the pH of running buffer decreased. The highest separation efficiency, largest peak area and most stable baseline were achieved when the pH was 2.5. Accordingly, the NaH_2_PO_4_ (HCl) solution at pH 2.5 was considered as the optimal running buffer for following studies.

To obtain a better resolution, the concentration of running buffer (pH 2.5) containing 1.0 mg/L histamine at 10 kV applied voltage needed to be optimized. Figure 2e reveals that the migration time of histamine decreased when the buffer concentration was increased from 50 mmol/L to 100 mmol/L and then slightly increased in the concentration range of 100–150 mmol/L. Accordingly, subsequent experiments were performed in a 100 mmol/L running buffer solution.

The CE separation voltage had a significant effect on the separation result of the histamine. The variations in peak area and migration time at separation voltages ranging from 6 kV to14 kV are shown in Figure 2f. Experiments used a 100 mmol/L running buffer (pH 2.5) containing 1.0 mg/L histamine. The resolution of histamine improved slightly as the separation voltage was increased from 6–10 kV. The separation became poor in the range of 10–14 kV with an unstable baseline and an asymmetric peak shape. Accordingly, the best result was obtained at the separation voltage of 10 kV, with this value being used in following experiments.

### 2.4. Analytical Parameters

A MISPE-CE method for the detection of histamine was established in this study. As shown in Figure 3a,b, the peak area gradually increased with the concentration of the histamine standard solution from 0.1 μg/L to 100.0 μg/L. Excellent linearity was obtained between the peak area and the concentrations of histamine. The linear equation was y = 7289x + 28,374 with R^2^ = 0.9982. The limit of detection (LOD) (S/N = 3) was calculated to be 0.087 μg/L, and the limit of quantitation (S/N = 10) was 0.29 μg/L. Compared with the slope of the standard curve without MISPE pre-treatment (Figure 3c,d), the enrichment factor of the MISPE-CE method was 14.7.

### 2.5. Repeatability, Selectivity and Anti-Interference Capability of the MISPE-CE Method

The results showed that the MIPSE-CE method exhibited good repeatability with the RSD of intraday and interday precision based on both peak area and migration time values being less than 6.6%. The RSD of CE peak area values for 9 MISPR-CE in a batch was less than 20.0%. Experiments on batch-to-batch repeatability were also conducted using two MISPE cartridges and the results showed that the RSD of CE peak area was less than 5.0%.

The CE peak areas obtained using MISPE cartridges and C_18_ cartridges as pre-treatment procedure were compared to assess the selectivity of the MISPE-CE method (Figure 4a). The CE peak area of the MISPE cartridges was about 3 times higher than that of C_18_ cartridges, indicating the good binding affinity of MISPE cartridges for histamine. Structural analogues of histamine (tryptamine and phenethylamine) were selected to further study the selectivity of MIP. Figure 4b shows that the CE peak area of the MIP was much higher than the NIP for histamine (A), but there was no significant difference between the MIP and NIP for tryptamine (B) and phenethylamine (C), indicating that the MIP had selective adsorption properties for histamine. Imprinting factors of the MIP for histamine, tryptamine and phenylethylamine were calculated to be 1.8, 1.0 and 0.9, respectively. These results demonstrate that MISPE cartridges were suitable for the selective detection histamine in foods.

The anti-interference capability was next investigated by adding possible coexisting substances which usually exist in real food samples to a 0.1 μg/L histamine standard solution, including a 5-fold concentration of amino acids (arginine and lysine), 100-fold concentration of glucose and 50-fold concentration of inorganic ions (Na^+^, K^+^, CH_3_COOH^−^ and Cl^−^). The detection results proved the RSD of the CE peak area ranged from 0.6% to 7.2% (Figure 4c). Accordingly, the anti-interference capability of the MISPE-CE was acceptable.

### 2.6. Accuracy and Applicability of the MISPE-CE Method

To explore the accuracy of the proposed method for the analysis of real samples, the level of histamine in spiked rice vinegar and liquor samples were determined using the MISPE-CE method. As shown in Table 1, the recoveries of histamine in the rice vinegar and liquor were in the range of 92.63–111.00% with the RSD of 2.03–4.26%. The results indicated that the MISPE-CE method possessed satisfactory accuracy.

The concentration of histamine in real samples including fish, prawn, pork, chicken breast and soy sauce were also determined by the HPLC and MISPE-CE method to evaluate the wider applicability of the method. The electropherograms for histamine detection in these different foods are depicted in Appendix A and the results are summarized in Appendix A. No significant difference was found between the MISPE-CE method and HPLC method (*p* > 0.05). Therefore, the MISPE-CE method was reliable for the detection of histamine in food samples.

### 2.7. Merits of the MISPE-CE Method

Compared with the methods previously reported to date for histamine detection in foods (Appendix A), the MISPE-CE method offered the widest linear range [34,35,36]. The LOD was also lower than most previous studies using CE for the determination of histamine [37,38]. More importantly, the sample pre-treatment was simplified because of the good selectivity of the MIP. Thus, the MISPE-CE method was a sensitive tool for the detection of histamine.

## 3. Materials and Method

### 3.1. Reagents and Materials

Rice vinegar, liquor, soy sauce, fish, prawn, pork and chicken breast samples were purchased from a local supermarket (Taian, China).

Histamine, tryptamine and phenylethylamine were purchased from Yuanye Biological Technology Co., Ltd. (Shanghai, China). Histamine dihydrochloride and ethylene glycol dimethacrylate were obtained from Aladdin Biochemical Technology Co., Ltd. (Shanghai, China). Methacrylic acid was provided by Macklin Biochemical Technology Co., Ltd. (Shanghai, China), and the 2,2′-azobis (2-methylpropionitrile) were supplied by West Asia Chemical Technology Co., Ltd. (Shandong, China). Methanol, ethanol, acetic acid, sodium dihydrogen phosphate, sodium acetate, sodium chloride, potassium chloride and glucose were purchased from Kaitong Chemical Reagent Co., Ltd. (Tianjin, China). Hydrochloric acid was provided by Kemiou Chemical Reagent Co., Ltd. (Tianjin, China). Sodium hydroxide was supplied by BASF Chemical Trade Co., Ltd. (Tianjing, China). Arginine and lysine were obtained from Solarbio Bioscience & Technology Co., Ltd. (Shanghai, China). Dansyl chloride and ammonium acetate were purchased from Aladdin Biochemical Technology Co., Ltd. (Shanghai, China). Trichloroacetic acid was provided by Damao Chemical Reagent Co., Ltd. (Tianjin, China). Sodium bicarbonate was obtained from Yongda Chemical Reagent Co., Ltd. (Tianjin, China). Sodium glutamate was supplied by Biotopped Life Sciences Co., Ltd. (Beijing, China). Diethyl ether was purchased from Yantai Yuandong Fine Chemical Co., Ltd. (Shandong, China). Acetonitrile, methanol and acetone were supplied by Yuwanghetianxia New Material Co., Ltd. (Shandong, China). The acetonitrile and methanol were chromatographic grades, and the purity of all other chemicals was >98%. Double distilled water (DDW) was prepared by the Aike ultrapure water instrument (Chengdu, China).

### 3.2. Apparatus and CE Conditions

The Fourier transform infrared (FT-IR) spectra (4000–500 cm^−1^) were obtained on a Nicolet iS10 infrared spectrometer (Thermo, USA). The HPLC system was equipped with two pumps (LC-20AT), a C_18_ reversed-phase column (4.6 × 250 mm, 5 μm) and a UV detector (SPD-20A, Shimadzu, Japan). Mobile phase A was the mixture of ammonium acetate (0.01 mol/L) and acetonitrile (10:90, *v*/*v*). Mobile phase B was the mixture of ammonium acetate (0.01 mol/L) and acetonitrile (90:10, *v*/*v*). The injection volume was 20 μL, the flow rate was 0.8 mL/min and the detection wavelength was 254 nm.

Histamine was separated using a Beckman P/ACE MDQ capillary electrophoresis system with a diode array detection (DAD) at 211 nm. Separation steps were carried out on uncoated fused-silica capillaries (Yongnian Optical Conductive Fiber Plant, Hebei, China) of 75 μm i.d.. The capillary was firstly thermostated at 25℃ and then successively flushed with 0.1 mol/L NaOH, DDW and the running buffer for 5 min each. Standard solutions and samples were injected by pressure at 0.5 psi for 5 s and separated under 10 kV with a positive voltage. The data were collected and processed by Beckman P/ACE 32 Karat software Version 8.0.

### 3.3. Synthesis of the MIP

The MIP and non-imprinted polymer (NIP) were synthesized by bulk polymerization using our previously reported method [39]. The synthetic procedure for the MIP is depicted in Figure 1a.

### 3.4. The MISPE-CE Procedure

The MISPE-CE procedure is shown in Figure 1b. Firstly, 200.0 mg MIP was packed into an empty MISPE cartridge. Subsequently, the cartridges were activated with 5.0 mL of methanol and DDW, followed by loading with 25.0 mL of a 100.0 μg/L histamine standard solution. Then, the MISPE cartridges were eluted with 8.0 mL of methanol-acetic acid (90:10, *v*/*v*). Finally, the eluents were dried under N_2_ at 40 °C, and then redissolved in 1.0 mL of running buffer for subsequent CE separation and analysis.

### 3.5. Sample Preparation

To verify the accuracy of the method, recovery tests were carried out. Spiked rice vinegar and liquor samples were prepared as follows. Firstly, 1.0 mL of the samples were transferred into 15.0 mL-centrifuge tubes and then spiked with 1.0 mL histamine standard solution at different levels of 8.0 μg/L, 10.0 μg/L and 15.0 μg/L. Secondly, 0.5 g of sodium chloride was added to each solution, and the mixtures shaken. Next, 5.0 mL of ether was added and vortexed until two layers formed, with the ether layer then being transferred to another 15.0 mL-centrifuge tube. Subsequently, the extraction processes were repeated twice. The extracts were collected and evaporated to dryness with N_2_ under a water bath at 40 °C. The residues were then redissolved in 25.0 mL of ethanol. Finally, the amount of histamine in the extracts was analyzed by the CE method.

The liquid samples (soy sauce) and meat samples (fish, prawn, pork and chicken breast) were prepared according to the national standard method of GB 5009.208-2016 (samples prepared for CE analysis without derivatization), and then analyzed by the HPLC and CE method separately.

## 4. Conclusions

A MISPE-CE method was developed for the enrichment and determination of histamine in foods. The method possessed high repeatability, selectivity and anti-interference capability, along with outstanding accuracy and applicability. This paper provides a promising strategy for the monitoring of histamine in foods.

## Data Availability

The date presented in this work are available in the article and Appendix A.

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
