# Peer review of "A Capillary Electrophoresis Method Based on Molecularly Imprinted Solid-Phase Extraction for Selective and Sensitive Detection of Histamine in Foods"

_molecules, 2022, doi:10.3390/molecules27206987_

Round 1

Reviewer 1 Report

In Figure 1, rather than using the peak at 1024/1029  cm-1, which is difficult to see, perhaps examine the peak at approx 700 cm-1, which is really strong for histamine and can be seen clearly in Figure 1 in the MIP before elution.  Given that tryptamine and phenylethylamine is immobilised on the cartridge ( Figure 4), I am surprised that the signal is so clean for such a complex sample as soy sauce in Figure S1 (a). otherwise the paper is solid, in that it compares MIP and NIP

Author Response

Response to Reviewer 1

Thank you very much for giving us an opportunity to revise our manuscript, we appreciate reviewers very much for your positive and constructive comments and suggestions. We have studied reviewer’s comments carefully and have made revision which marked in red in the paper. We have tried our best to revise our manuscript according to the suggestions. To sum up, the following revisions are made:

Point 1: In Figure 1, rather than using the peak at 1024/1029 cm-1, which is difficult to see, perhaps examine the peak at approx. 700 cm-1, which is really strong for histamine and can be seen clearly in Figure 1 in the MIP before elution. Given that tryptamine and phenylethylamine is immobilised on the cartridge (Figure 4), I am surprised that the signal is so clean for such a complex sample as soy sauce in Figure S1 (a). otherwise, the paper is solid, in that it compares MIP and NIP.

Response 1: Thank you very much for your good advice. We have examined the peak at approx. 700 cm-1, and the absorption peak at 617 cm-1 in histamine dihydrochloride and 622 cm-1 in the MIP before elution were observed. The manuscript and Figure 1 have been revised. Please see the revised manuscript page 5, lines 77-78.

Reviewer 2 Report

The paper “A capillary electrophoresis method based on molecularly imprinted solid-phase extraction for selective and sensitive detection of histamine in foods“ by Yixuan Fan, Runze Yu, Yongfeng Chen, Yufeng Sun, Geoffrey I.N. Waterhouse and Zhixiang Xu describes analytical developments of some practical interest. The authors combine well known SPE sample preparation based on MIPs with CE. Generally, CE suffers on a limited sensitivity. This is compensated for  by an efficient sample pretreatment by SPE-MIPs. Thus, a wider range of linear detection resulted. Interferences were also considered which were not predominant due to the high separation efficiency of CE and sample preparation.  All experiments were thoroughly described and discussed. This refers to chromatographic activities as eluent optimization and e.g. repeatability of measurements. Thus, the paper can be published as it is.

Author Response

Response to Reviewer 2

Thank you very much for giving us an opportunity to revise our manuscript, we appreciate reviewers very much for your positive and constructive comments and suggestions. We have studied reviewer’s comments carefully and have made revision which marked in red in the paper. We have tried our best to revise our manuscript according to the suggestions. To sum up, the following revisions are made:

Point 1: The paper “A capillary electrophoresis method based on molecularly imprinted solid-phase extraction for selective and sensitive detection of histamine in foods” by Yixuan Fan, Runze Yu, Yongfeng Chen, Yufeng Sun, Geoffrey I.N. Waterhouse and Zhixiang Xu describes analytical developments of some practical interest. The authors combine well known SPE sample preparation based on MIPs with CE. Generally, CE suffers on a limited sensitivity. This is compensated for by an efficient sample pretreatment by SPE-MIPs. Thus, a wider range of linear detection resulted. Interferences were also considered which were not predominant due to the high separation efficiency of CE and sample preparation. All experiments were thoroughly described and discussed. This refers to chromatographic activities as eluent optimization and e.g. repeatability of measurements. Thus, the paper can be published as it is.

Response 1: Thank you very much for your high appraisement on our manuscript.

Reviewer 3 Report

In this work the authors propose a MISPE-CE method for the analysis of in different real samples.

Some following issues should be carefully considered to further improve the quality of this paper:

·         In the abstract: The limit of detection and limit of quantification of the method were calculated. The last phrase should be rephrased.

·         Line 35: “it suffers from time-consuming and unstable of antibody” – must be rephrased

·         Lines 55-56: italics must be used for Latin names of plants

·         In the introduction, there is no discussion of other reported histamine-imprinted polymers, even if they were employed in other analytical techniques.

·         Scheme 1: Template removal instead of Remove template

·         In section: 2.2. Optimization of the MISPE Procedures, what represents the 100.0 μg/L histamine solution should be explained in a more detailed manner. This was the solution used in loading step of the SPE analysis?

·         Line 110: higher separation efficiency

·         In section: 2.3. Optimization of the CE Conditions, in text or in Figure 2, should be presented all CE conditions at which the separations were realized. For example, when different pHs where tested, what potential, buffer concentration, etc. were used?

·         The following phrase is unclear: The separation voltage increased rapidly, but the resolution of histamine improved only slightly. A 2 kV rise in the applied potential is not such a big increase.

·         And the best result was obtained at the separation voltage of 10 kV. There is no explanation why 10 kV potential was selected. What was the monitored parameter? Shorter migration time or a better resolution? However, it cannot be evaluated the separation efficiency only using a standard solution histamine.

·         Figure 3b and d: CE absorbance of the histamine should be rephrased. For example: Electropherograms showing the analysis/separation of histamine…

·         Figure 3d: it looks like histamine signal at 10 mg/L it’s saturated because of the poor peak shape, however this is strange because 10 μg/mL is still a low concentration for CZE.

·         Line 148: The CE peak area of the MISPE cartridges was about 3 times higher than that of C18 cartridges, indicating the good selectivity of MISPE cartridges toward the histamine. – The selectivity is evaluated by comparing the MIP response towards template and other structural analogues. Here a good binding affinity can be used instead of selectivity.

·         Line 152: indicating that MIP had stronger CE absorbance to histamine. A polymer cannot have a stronger CE absorbance to an analyte. Good adsorption properties for the analyte can be used.

·         In table S2, for comparative purposes, the units of measure should be the same.

·         The analysis of histamine in real samples by HPLC was also realized by combining it with MISPE?

The manuscript is superficially written and in many places the sentences should be rephrased. The methods and the results presented are unclear and not explicitly presented. The paper gives some useful data, but the paper's writing and structure are not well done. Also, because of the poor English, the manuscript should be modified by a native speaker.

In conclusion, this manuscript is not suitable for publication in the Molecules journal.

Author Response

Response to Reviewer 3

Thank you very much for giving us an opportunity to revise our manuscript, we appreciate reviewers very much for your positive and constructive comments and suggestions. We have studied reviewer’s comments carefully and have made revision which marked in red in the paper. We have tried our best to revise our manuscript according to the suggestions. To sum up, the following revisions are made:

Point 1: In the abstract: The limit of detection and limit of quantification of the method were calculated. The last phrase should be rephrased.

Response 1: We are sorry to make this mistake. The sentence of “The limit of detection and limit of quantification of the method were calculated” has been modified. Please see the revised manuscript page 2, line 20.

The last phrase of the abstract has been revised carefully. Please see the revised manuscript page 2, lines 22-25.

Point 2: Line 35: “it suffers from time-consuming and unstable of antibody” – must be rephrased.

Response 2: Thanks for this good advice. The article has been revised. Please see the revised manuscript page 3, line 39.

Point 3: Lines 55-56: italics must be used for Latin names of plants.

Response 3: We are very sorry for our negligence and the Latin names of plants have been added, please see the revised manuscript page 4, line 61.

Point 4: In the introduction, there is no discussion of other reported histamine-imprinted polymers, even if they were employed in other analytical techniques.

Response 4: Thanks for this good advice. The discussion of other reported histamine-imprinted polymers has been added in the introduction. Please see the revised manuscript page 4, lines 51-55.

Point 5: Scheme 1: Template removal instead of Remove template.

Response 5: We are sorry for our incorrect writing. The Scheme 1 has been corrected. Please see the revised Scheme 1.

Point 6: In section: 2.2. Optimization of the MISPE Procedures, what represents the 100.0 μg/L histamine solution should be explained in a more detailed manner. This was the solution used in loading step of the SPE analysis?

Response 6: Thanks for this good advice. The 100.0 μg/L histamine solution used in loading step has been explained in section: 3.4. The Procedure of MISPE-CE. Please see the revised manuscript page 11, line 215.

Point 7: Line 110: higher separation efficiency.

Response 7: Thanks for this good advice. The “higher separation efficient” has been changed to “higher separation efficiency”. Please see the revised manuscript page 6, lines 105-106.

Point 8: In section: 2.3. Optimization of the CE Conditions, in text or in Figure 2, should be presented all CE conditions at which the separations were realized. For example, when different pHs where tested, what potential, buffer concentration, etc. were used?

Response 8: Thanks for this good advice. All CE conditions have been added. Please see the revised manuscript section 2.3. and the Figure Captions.

Point 9: The following phrase is unclear: The separation voltage increased rapidly, but the resolution of histamine improved only slightly. A 2 kV rise in the applied potential is not such a big increase.

Response 9: Thanks for this good advice. This sentence has been checked and revised carefully. Please see the revised manuscript page 7, lines 118-120.

Point 10: And the best result was obtained at the separation voltage of 10 kV. There is no explanation why 10 kV potential was selected. What was the monitored parameter? Shorter migration time or a better resolution? However, it cannot be evaluated the separation efficiency only using a standard solution histamine.

Response 10: Thanks for this good advice. The resolution of histamine improved slightly with the separation voltage increasing from 6-10 kV. The separation result became bad in the range of 10-14kV with unstable baseline and asymmetric peak. Thus, the best result was obtained at the separation voltage of 10kV.

The shorter migration time and better resolution were both the monitored parameters in this study.

It is true that it cannot be evaluated the separation efficiency only using a standard solution histamine. This sentence has been revised carefully. The “separation efficiency” has been changed to “separation result”. Please see the revised manuscript page 6, line 115.

Point 11: Figure 3b and d: CE absorbance of the histamine should be rephrased. For example: Electropherograms showing the analysis/separation of histamine.

Response 11: Thanks for this good advice. Please see the revised manuscript page 20, line 369 and 371.

Point 12: Figure 3d: it looks like histamine signal at 10 mg/L it’s saturated because of the poor peak shape, however this is strange because 10 μg/mL is still a low concentration for CZE.

Response 12: Thanks for this good advice. We are sorry to make this mistake. This experiment has been repeated, and the Figure 3d has been revised. Please see the revised Figure 3d.

Point 13: Line 148: The CE peak area of the MISPE cartridges was about 3 times higher than that of C18 cartridges, indicating the good selectivity of MISPE cartridges toward the histamine. – The selectivity is evaluated by comparing the MIP response towards template and other structural analogues. Here a good binding affinity can be used instead of selectivity.

Response 13: Thanks for this advice. This sentence has been revised. Please see the revised manuscript page 8, lines 138-139.

Point 14: Line 152: indicating that MIP had stronger CE absorbance to histamine. A polymer cannot have a stronger CE absorbance to an analyte. Good adsorption properties for the analyte can be used.

Response 14: Thanks for this advice. This sentence has been revised. Please see the revised manuscript page 8, line 143.

Point 15: In table S2, for comparative purposes, the units of measure should be the same.

Response 15: Thanks for this good advice. The units of measure have been changed. Please see the revised supplementary materials Table S2.

Point 16: The analysis of histamine in real samples by HPLC was also realized by combining it with MISPE?

Response 16: The analysis of histamine in real samples by HPLC was realized according to the national standard method of GB 5009.208-2016 without MISPE. Please see the section: 3.5. Sample Preparation.

Point 17: The manuscript is superficially written and in many places the sentences should be rephrased. The methods and the results presented are unclear and not explicitly presented. The paper gives some useful data, but the paper's writing and structure are not well done. Also, because of the poor English, the manuscript should be modified by a native speaker.

Response 17: Thanks for your good advice. Our manuscript has been checked and revised carefully. Please see the revised manuscript.

Round 2

Reviewer 3 Report

The authors have resolved some specific issues, however there are still substantial problems in the methods and results sections. The content of the experimental section is recommended to be expanded with some new experiments. In my opinion, I don't think this work meets the publishing requirements of Molecules journal, both in terms of experimental contents and innovation.